# De-biasing Weakly Supervised Learning by Regularizing Prediction Entropy

**Dean Wyatte**
Activision
dean.wyatte@activision.com

## Abstract

We explore the effect of regularizing prediction entropy in a weakly supervised setting with inexact class labels. When underlying data distributions are biased toward a specific subclass, we hypothesize that entropy regularization can be used to bootstrap a training set that mitigates this bias. We conduct experiments over multiple datasets under supervision of an oracle and in a semi-supervised setting finding substantial reductions in training set bias capable of decreasing test error rate. These findings suggest entropy regularization as a promising approach to de-biasing weakly supervised learning systems.

## 1 Introduction

Weakly supervised learning is often encountered in real-world settings where it is infeasible to collect massive training sets with detailed annotations. For example, machine learning systems may start with a relatively small dataset of hand-labeled examples with the goal of bootstrapping a larger dataset over time (Reed et al., 2014). Due to the high cost of labeling data, class labels may be inexact with multiple underlying data distributions (Zhou, 2017). Care should be taken to account for the underlying mixture when dealing with inexact labels as failure to do so could introduce bias leading to covariate shift (Shimodaira, 2000) when adding new examples to the training set.

Weak supervision is often supplemented with a human-in-the-loop serving as an oracle that provides labels for only the most informative examples (Settles, 2009). However, the cost of labeling informative examples may directly compete with other costs such as human verification of model predictions. Motivated by the observation that regularizing prediction entropy during training increases diversity in top-ranking predictions (Figure 1), we propose entropy regularization as a simple but effective approach for satisfying these multiple costs and de-biasing weakly supervised learning systems with inexact labels.

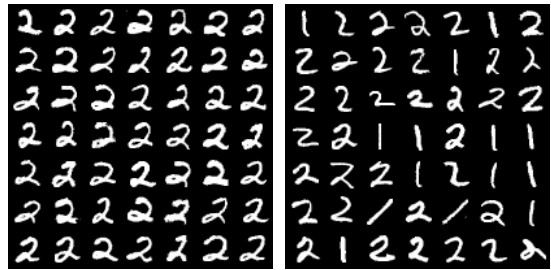

Figure 1: Examples from a biased mixture class composed of MNIST digits *2* and *1* with sampling rates 75% and 25% ranked by prediction probability. Standard training produces a biased ranking (left) whereas regularizing prediction entropy produces a substantially more diverse ranking (right).

Using prediction entropy as a regularizer has been shown to improve generalization across a range of tasks and models (Pereyra et al., 2017) and is especially effective when training data are limited (Dubey et al., 2017).

The entropy $H$ of a conditional probability distribution $p_\theta(\boldsymbol{y}|\boldsymbol{x})$ is given by:

$$H(p_\theta(\boldsymbol{y}|\boldsymbol{x})) = -\sum_i p_\theta(\boldsymbol{y}_i|\boldsymbol{x}) \log(p_\theta(\boldsymbol{y}_i|\boldsymbol{x})) \qquad (1)$$

Prediction entropy can be regularized by subtracting it from the negative log-likelihood:

$$\mathcal{L}(\theta) = -\sum \log p_\theta(\boldsymbol{y}|\boldsymbol{x}) - \beta H(p_\theta(\boldsymbol{y}|\boldsymbol{x})) \qquad (2)$$

with $\beta$ controlling the relative strength of the regularization, promoting maximum entropy predictions while satisfying the primary objective.

The result is a more general representation (Dubey et al., 2017) that produces prediction probabilities with a rank ordering robust to bias. This is not the same as simply ranking examples by entropy (e.g., uncertainty sampling, Settles, 2009) as the proposed ranking can be conditioned on a specific class with maximum probability.

## 2 EXPERIMENTS

To evaluate the de-biasing effect of entropy regularization, we created controlled datasets where the bias present in each class could be systematically manipulated. We used the original classes of each dataset to randomly create new mixture classes with a sampling rate of 75% for the biasing class (medium bias) or 95% for the biasing class (extreme bias).

For each experiment below, we start by defining the mapping of original classes to mixture classes and then split the original training set of 60000 examples into two pools of 50000 examples and 10000 examples. We randomly sample the initial training set from the larger pool of 50000 examples according to the specified biased sampling rate. The smaller pool of 10000 examples remains unbiased, that is, is sampled with a rate of 100% for each underlying class. We train various neural network architectures using the initial biased training set and then proceed to rank examples from the pool of 10000 unbiased examples by each $p(class_i)$, selecting the top $n$ examples from the ranking to add to the training set. Labels are provided by an oracle resembling a human-in-the-loop or inferred in a semi-supervised manner. This procedure is repeated for 10 rounds during which we monitor the bias of each class in the growing training set and the error rate on the original dataset's test set with the unbiased class mapping applied. Model weights are initialized at the start of each round.

All training used the Adam optimizer (Kingma & Ba, 2014). The initial learning rate and any learning rate schedules were set using an unbiased mixture dataset for one round of training without entropy regularization. The weight of the entropy regularization term ($\beta$ in Equation 2) was fixed at 1.0 when used during training. All experiments included a small amount of data augmentation (random crops with a padding of four pixels and random horizontal flips in the case of CIFAR-10 and CIFAR-100). Models were implemented in PyTorch (Paszke et al., 2017) and trained on NVIDIA RTX 2080 Ti GPUs.

### 2.1 MNIST

As a preliminary experiment, we trained fully-connected networks with two hidden layers of 1024 ReLU units on a mixture dataset of MNIST digits (LeCun et al., 1998) using a fixed learning rate of 0.0001 for 50 epochs. The mixture dataset was composed of five disjoint classes, each containing two random MNIST classes. At the end of each round, the training set was updated with the top 100 examples from each class ranked by prediction probability with any labels provided by an oracle.

We observed a substantial decrease in training set bias across rounds of updates for networks trained with entropy regularization with an overall decrease of 5%-10% after 10 rounds (Figure 2). Results are reported in Table 1.

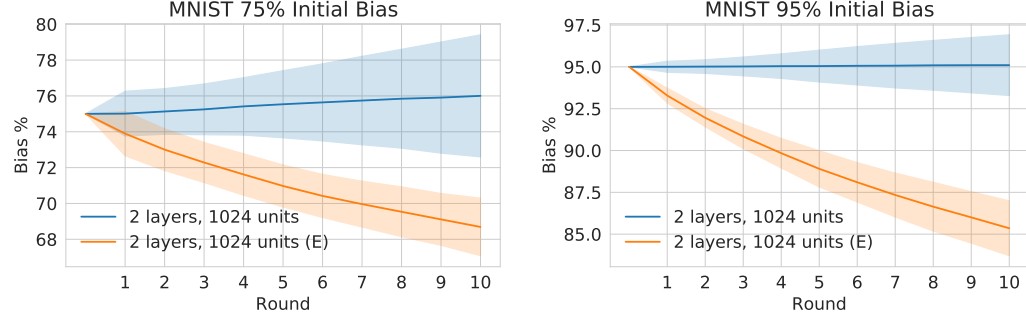

Figure 2: MNIST mixture bias over 10 rounds of updates for 75% initial bias (left) and 95% initial bias (right). Shaded regions indicate standard deviation over five runs. E: Entropy Regularization.

Table 1: MNIST mixture bias after 10 rounds of updates. E: Entropy Regularization.

| METHOD | 75% INITIAL BIAS | 95% INITIAL BIAS |
|---|---|---|
| 2 layers, 1024 units | $76.00 \pm 3.40$ | $95.10 \pm 1.81$ |
| 2 layers, 1024 units (E) | $68.69 \pm 1.60$ | $85.35 \pm 1.64$ |

## 2.2 CIFAR-10 AND CIFAR-100

We repeated the above experiment with the CIFAR-10 and CIFAR-100 datasets (Krizhevsky & Hinton, 2009) to evaluate the effect of entropy regularization on more challenging biased classification tasks. For CIFAR-10 mixtures, we again created five disjoint classes, each containing two random CIFAR-10 classes. For CIFAR-100 mixtures, we used the provided coarse labels for grouping the original classes into 20 superclasses, randomly choosing one class from each superclass to be the biasing class sampled at a rate of 75% or 95% with the remaining samples spread evenly across the other four contributing classes (i.e., 6.25% each in the 75% bias case and 1.25% each in the 95% bias case). At the end of each round, the training set was updated with the top 100 examples from each class ranked by prediction probability in the case of CIFAR-10 and the top 10 examples from each class in the case of CIFAR-100 due to it having fewer examples per class. Labels were provided by an oracle.

We trained the 20-layer ResNet variant from He et al. (2016) and the 40-layer DenseNet variant from Huang et al. (2017) on these tasks. The DenseNet implementation used bottleneck and transition layers with a compression rate of 0.5 (Densenet-BC in the original formulation). The networks were trained with an initial learning rate of 0.001 for 100 epochs in the case of CIFAR-10 and 120 epochs in the case of CIFAR-100. The learning rate was reduced by a factor of 10 after 50% and 75% of the total epochs.

Again, we observed a decrease of 5%-10% in training set bias after 10 rounds of updates for networks trained with entropy regularization (Table 2).

Table 2: CIFAR-10 and CIFAR-100 mixture bias after 10 rounds of updates. E: Entropy Regularization.

| METHOD | 75% INITIAL BIAS | | 95% INITIAL BIAS | |
|---|---|---|---|---|
| | CIFAR-10 | CIFAR-100 | CIFAR-10 | CIFAR-100 |
| ResNet20 | $72.10 \pm 3.12$ | $72.65 \pm 4.47$ | $93.80 \pm 2.04$ | $91.14 \pm 3.99$ |
| ResNet20 (E) | $68.45 \pm 1.66$ | $67.18 \pm 2.18$ | $83.77 \pm 1.55$ | $84.69 \pm 2.81$ |
| DenseNet40 | $73.09 \pm 3.06$ | $72.40 \pm 4.46$ | $94.23 \pm 1.77$ | $90.76 \pm 3.84$ |
| DenseNet40 (E) | $68.69 \pm 1.56$ | $67.18 \pm 2.03$ | $84.00 \pm 1.67$ | $84.71 \pm 2.36$ |

## 2.3 SEMI-SUPERVISED LEARNING

To evaluate whether entropy regularization could be useful in semi-supervised settings lacking an oracle, we repeated our CIFAR-10 and CIFAR-100 experiments assigning the label of the class with the highest predicted probability to each example when updating the training set between rounds (Lee, 2013). Here, we monitor the error rate on each original dataset's test set with the unbiased class mapping applied.

Figure 3 shows the effect of entropy regularization on CIFAR-10 mixture test error rate across rounds of training set updates under extreme bias. The substantial decrease in training set bias across rounds of updates produces a proportional decrease in test error rate. This effect is most pronounced when labels for training set updates are provided by an oracle, but is also apparent in a semi-supervised setting suggesting that the updates are beneficial while maintaining low label noise. This same effect was unreliable in our CIFAR-100 experiments, likely due to the limited number of examples per class compared to CIFAR-10 and unexplored hyperparameter optimization. Full results are reported in Table 3.

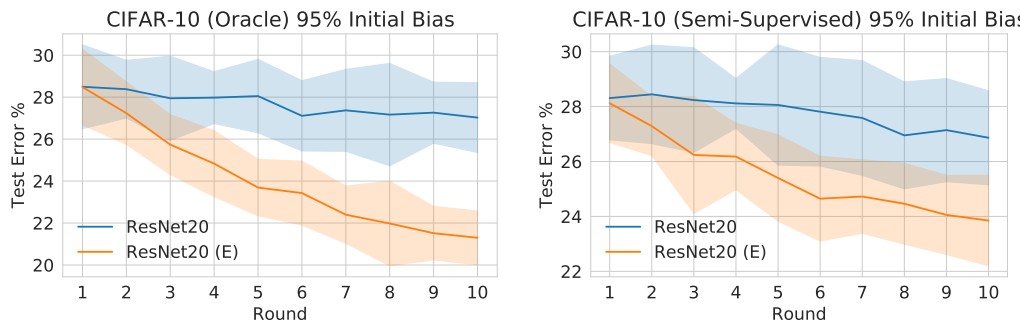

Figure 3: CIFAR-10 mixture test error rate over 10 rounds of updates with 95% initial bias for oracle (left) and semi-supervised (right) experiments. Shaded regions indicate standard deviation over five runs. E: Entropy Regularization.

Table 3: CIFAR-10 and CIFAR-100 mixture test error rate after 10 rounds of updates with 95% initial bias. E: Entropy Regularization.

| METHOD | ORACLE | | SEMI-SUPERVISED | |
| --- | --- | --- | --- | --- |
| | CIFAR-10 | CIFAR-100 | CIFAR-10 | CIFAR-100 |
| ResNet20 | $27.03 \pm 1.66$ | $60.57 \pm 0.99$ | $26.87 \pm 1.71$ | $61.54 \pm 1.45$ |
| ResNet20 (E) | $21.30 \pm 1.28$ | $58.51 \pm 1.48$ | $23.85 \pm 1.64$ | $60.65 \pm 1.38$ |
| DenseNet40 | $27.58 \pm 1.33$ | $60.56 \pm 1.11$ | $28.27 \pm 2.01$ | $60.92 \pm 0.86$ |
| DenseNet40 (E) | $22.10 \pm 1.11$ | $58.47 \pm 0.75$ | $24.68 \pm 1.06$ | $60.36 \pm 0.73$ |

## 3 CONCLUSION

We conducted a range of experiments in a weakly supervised setting where the bias present in inexact class labels with multiple underlying data distributions could be manipulated. Our results indicate that regularizing prediction entropy has a de-biasing effect for top-ranking predictions that can be exploited to bootstrap training sets from an unbiased pool of examples and incrementally decrease error.

Our approach is particularly effective when an oracle is available to provide correct labels, but could also be viable in semi-supervised settings where labels are inferred from prediction probabilities. Overall, we find entropy regularization to be a promising approach for de-biasing weakly supervised learning systems.

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
