# OpenReview forum: "De-biasing Weakly Supervised Learning by Regularizing Prediction Entropy"
_ICLR.cc/2019/Workshop/LLD — LLD 2019_

### Official Review · AnonReviewer2 · 2019-03-30
**Promising use of entropy penalty to de-bias noisy datasets, but context could be made clearer**

**Rating:** 3
**Confidence:** 2

**Review:**

Summary: The paper proposes to use the well-known technique of entropy regularization to de-bias the training set. The classes in the training set are assumed to come from biased mixtures of true classes. By penalizing low-entropy prediction of the model, combined with new labels from an oracle or inferred from prediction probabilities, the procedure can reduce the class bias in the noisy dataset. Experiments on MNIST, CIFAR-10 and CIFAR-100 validates the utility of the proposed scheme, showing that the bias decreases after the procedure.

Strengths:
1. The use of entropy penalty to de-bias noisy dataset is novel to the best of my knowledge.
2. Experimental results on MNIST, CIFAR-10, and CIFAR-100 are convincing and validate the claim in the paper.

Weaknesses:
1. Bias is never defined. I take it to mean class imbalance in the mixture class. I recommending elaborating on the set up and the context where this setting makes sense.
2. The experimental procedure isn't clear. What are mixture classes? Why does it make sense to construct mixture classes as such? What kind of real scenario are these experiments simulating?

---

### Official Review · AnonReviewer1 · 2019-04-05
**Review: De-biasing Weakly Supervised Learning by Regularizing Prediction Entropy**

**Rating:** 5
**Confidence:** 2

**Review:**

The authors propose using conditional entropy regularization during the training in semi-supervised settings to mitigate the bias of imbalanced data. The idea is elegant and effectively communicated, and the authors demonstrate its effectiveness empirically on MNIST and CIFAR-10/100.

I suggest the committee consider this submission for best paper.

I'm curious how the weight on the regularization term affects the bias mitigation.

Minor grammatical errors: "care should be taken *into* account", "a a substantially more diverse ranking".

---

### Decision · Program_Chairs · 2019-04-08
**Acceptance Decision**

Accept